# Specific binding of human P[28] rotavirus VP8* protein to blood group ABH antigens on type 1 chains

Yi Zheng[1,2☯], Xiaoman Sun[3☯], Yuting Li[2], Beibei Huang[2], Yang Chen[1,2], Han Zhou[1,2], Cuiyan Cao[1,2], Wengang Chai[4], Zhaojun Duan[3], Dandi Li[3*], Jingyu Yan🆔[1,2*], Xinmiao Liang[1,2*]

1 State Key Laboratory of Phytochemistry and Natural Medicines, Dalian Institute of Chemical Physics, Chinese Academy of Sciences, Dalian, China, 2 Ganjiang Chinese Medicine Innovation Center, Nanchang, China, 3 National Key Laboratory of Intelligent Tracking and Forecasting for Infectious Diseases (NITFID), NHC Key Laboratory for Medical Virology and Viral Diseases, National Institute for Viral Disease Control and Prevention, Chinese Center for Disease Control and Prevention, Beijing, China, 4 Glycosciences Laboratory, Faculty of Medicine, Imperial College London, Hammersmith, London, United Kingdom

☯ These authors contributed equally to this work.
* dandili@126.com (DL); yanjingyu@dicp.ac.cn (JY); liangxm@dicp.ac.cn (XL)

## Abstract

Group A rotavirus (RV) has been the major cause of acute gastroenteritis in infants and young children. Among the five P genogroups almost all P genotype RVs in P[II], P[III] and P[IV] genogroups that infect humans can bind glycan histo-blood group antigens (HBGAs) as the receptors on the host cell surface to infect host through the viral spike protein VP8*. Although P[I] is the largest genogroup, P[28] and P[10] are the only two genotype RVs infecting humans in the group. It has recently been found that a P[28] strain is related to bat RV and considered a possible product of reassortment between bat and human RVs. Bats are increasingly being recognized as an important reservoir for viruses crossing species barriers to infect humans. Unrevealing the interactions between RVs and host receptors is important for understanding RV evolution, infection, and epidemic. In the present study, using a multiphasic approach, including X-ray crystallography, glycan microarray with a dedicated probe library, bio-layer interferometry, site-specific mutagenesis, and molecular docking and dynamics simulations, we found that P[28]-VP8* can bind to all blood group A, B and H(O) antigens but on type 1 chain only, without the capability to bind to any Lewis epitopes or mucin O-glycan cores. Different from most of the prevalent human RVs, such as P[8], P[4] and P[6], the broad HBGA binding specificity of P[28]-VP8* and the fact of the recently identified a possible reassortment P[28] strain of bat and human RVs have raised the concern of a future possibility of P[I] genogroup RV epidemic. RV surveillance may also need to take the P[I] genogroup RVs into account in the future.

**Data availability statement:** The coordinates and structure factors for the P[28]-VP8* structures are deposited in the Protein Data Bank under accession numbers 9JAA (apo) and 9JAK (with LNFP I). All other relevant data are within the manuscript and its Supporting Information file.

**Funding:** This work was supported by the National Natural Science Foundation of China (NSFC) (No. 21934005 to XL and No. 22074143 to JY), Jiangxi Provincial Natural Science Foundation (No. 20232BAB215021 and 20242BAB25354 to YZ) and Jiangxi "Double Thousand Plan" (to JY), and in part by the Wellcome Trust Biomedical Resource grant (218304/Z/19/Z to WC). The funders had no role in study design, data collection and analysis, decision to publish, or preparation of the manuscript.

**Competing interests:** The authors have declared that no competing interests exist.

## Author summary

Rotaviruses (RVs) are the main cause of severe diarrhea in young children. Glycans have been documented as the major receptors for many RVs of the P[II], P[III] and P[IV] genogroups, but we know very little about the glycan receptors of P[I] genogroup RVs. In this study, we used glycan microarray, bio-layer interferometry and X-ray crystallography to investigate the structure details responsible for the interactions of the VP8* spike protein of P[28] RV with type 1 blood group ABH(O) antigens. P[28] has been reported as one of the two genotype viruses in the P[I] group to infect humans. The broad histo-blood group antigens binding specificity and the recently found a possible P[28] reassortment strain of bat and human RVs may indicate a possibility of P[I] group RV epidemic and necessity of future RV surveillance.

## Introduction

Group A rotavirus (RV) has been the major cause of acute gastroenteritis in infants and children under the age of five. There is a vast genetic and strain diversity of RVs, and RVs are classified into G and P genotypes based on the viral surface structural proteins VP7 and VP4, respectively. The P genotypes are further classified into five genogroups [1], P[I] to P[V], based on the sequences of the spike protein VP8*, one of the two proteolytic fragments of VP4 [2]. It has been found that almost all P genotype RVs in P[II], P[III] and P[IV] groups that commonly infect humans recognize histo-blood group antigens (HBGAs) of the host through VP8* in a genotype dependent manner. For example, the P[8] and P[4] RVs that are most prevalent strains infecting infants can bind to blood group H epitope on type 1 chain (H-T1) and Lewis HBGAs [3–6]. Human P[6] and P[19] RVs bind to H-T1 in addition to mucin cores [7,8], whereas human P[9], P[14] and P[25] RVs in P[III] genogroup, recognize blood group A antigen only [1,9]. Human P[11] RVs of the P[IV] group can bind to both type 1 and type 2 precursors of the HBGAs [10–12].

There have been at least 58 P genotypes identified so far (https://rega.kuleuven.be/cev/viralmetagenomics/virus-classification/rcwg). P[I] genogroup is the largest RV family and it contains more than 26 P genotypes but there is only a total of eight P genotypes in the P[II], P[III], and P[IV] groups [1,13]. While most RVs in P[I] group infect animals, only two P[I] RVs have been reported to infect humans [1,14]. Despite the large number of P genotypes in this genogroup, there have been only a few reports describing their recognition to glycan receptors. The glycan bindings of animal strains of P[1], P[2], P[3] and P[7] are reported to be sialic acid-dependent [15–20]. P[1] Nebraska calf diarrhea virus (NCDV) and P[2] simian SA11 strains were found to bind to gangliosides with a preference of the N-glycolyl form of sialic acid (Neu5Gc) over the N-acetyl form (Neu5Ac) [15], while P[7] porcine OSU [16] and CRW-8 [17] strains have similar affinities for monosialo-gangliosides with either Neu5Ac or Neu5Gc. Canine K9 [18] and P[3] rhesus RRV strains were reported to interact with Neu5Ac monosaccharide [19,20].

P[10] and P[28] are the only two human RVs in the P[I] group but very limited data are available on their potential glycan receptors. P[10]-VP8* has been shown to bind all ABO secretors and non-secretors in saliva-based ELISA assays, whereas the same strain can only recognize mucin O-glycan core 2 trisaccharide in glycan binding assay [21]. P[28] RV was first identified in Ecuador [22] and considered to infect human exclusively [1,14]. In a recent work based on glycan microarray analysis [14], it was reported that P[28]-VP8* of the Ecuador strain Ecu534 bound unusually to αGal trisaccharide and was considered as an αGal-binding protein although it could also recognize H type 1 precursors.

P[28] is not a prevalent genotype with no current epidemiological significance. However, Esona and colleagues have conducted a comprehensive molecular characterization of human P[28] Suriname strain, and found that this human strain has multiple genes related to bat RVs and considered that the strain is possibly a product of reassortment between bat and human RVs [23]. Bats are increasingly being recognized as an important reservoir host for viruses crossing species barriers to infect humans [24]. The bats related zoonotic risk found in rotaviruses has raised the concern of potential cross-species transmission and may require continued RV strain surveillance. A better understanding of P[28] has become important.

The Ecu534 strain shares a similar VP4 with the Surinamese strain with an amino acid similarity of 97.5% [23]. To have a better understanding of P[28] RV's glycan receptors, in the present study we employed a multiphasic approach, including X-ray crystallography, glycan binding analysis with a dedicated glycan microarray, bio-layer interferometry, site-specific mutagenesis, and molecular docking and dynamics simulations, to investigate in detail the nature and specificity of P[28]-VP8* derived from the Ecu534 strain.

## Results

### Crystal structure of P[28]-VP8*

To investigate the structural features of P[28]-VP8*, a recombinant P[28]-VP8* variant (62–232 amino acids) was expressed in *E. coli* and purified by affinity chromatography with GSH-agarose. The N-terminal GST tag was removed by thrombin cleavage to obtain a purified protein with the molecular weight of ~19 kDa (S1A Fig). The additional bands at ~26 kDa are from the GST tag, which has no impact on the protein interaction with glycan receptors (S1B Fig). The structure of P[28]-VP8* determined at the resolution of 1.28 Ångström (Å) (PDB ID: 9JAA, Table 1) belongs to the tetragonal space group $P2_12_12_1$ with one monomer in the asymmetric unit. P[28]-VP8* showed the typical galectin-like fold with two twisted β-sheets consisting of strands A, L, C, D, G, H and M, B, I, J, K, respectively (Fig 1A). According to the structural superimposition, P[28]-VP8* most closely resembles the P[7]-VP8*, with a RMSD value of 0.442 Å; it also resembles P[14]- and P[9]-VP8* with RMSD values of 0.483 Å and 0.532 Å (Fig 1B), respectively. The RMSD values partly reflect the similarity of the structures although do not have absolute correlation with the phylogenetic analysis of the sequences. Based on the previous report [13], the structure of P[7]-VP8* gave closer values to that of P[9]- and P[14]-VP8*s, similar to that of P[28]-VP8*. For the P[5]-VP8*, although it belongs to the P[I] genogroup, it was found that P[5]-VP8* structure was most similar that of P[8]-VP8* of P[II] genogroup [25]. Here, it is shown that the structural differences between P[28]-VP8* and P[5] [25] are more pronounced than those of P[9]- and P[14]-VP8*s.

### Glycan receptors of P[28]-VP8* and binding kinetics

A dedicated microarray containing 58 glycan probes was used for screening analysis of glycan receptors of the recombinant P[28]-VP8*. As shown in S1 Table, the panel of the glycan probes contains previously reported diverse sequences recognized by RV-VP8*, including blood group A, B and H [4,5] on type 1 (T1), type 2 (T2) and globo backbones with different chain lengths (e.g., tetra- to octasaccharide); Lewis A, B, X and Y (Le^a, Le^b, Le^x, and Le^y, respectively) [6]; Thr-linked mucin O-glycan cores 1−4 [7]; mono-, di- and trisialylated gangliosides [15,17]; variously sialylated or fucosylated glycans [13,26]; and T1 and T2 terminating backbone sequences, the so-called HBGA precursors [10,12], with different

**Table 1. Diffraction data collection and refinement statistics.**

| PDB code | 9JAA | 9JAK |
|---|---|---|
| Ligand | None | H-T1 pentasaccharide (LNFP I) |
| Resolution (Å) | 34.37-1.35 (1.28-1.28) | 50.86-1.29 (1.38-1.29) |
| Space group | $P2_12_12_1$ | $P2_12_12_1$ |
| Unit cell parameters (a,b,c)(Å), (α,β,γ)(°) | (36.68, 41.33, 98.42), (90.0, 90.0, 90.0) | (36.18, 42.66, 101.72), (90.0, 90.0, 90.0) |
| No. of measured reflections | 329216 (34430) | 171770 (2690) |
| No. of unique reflections | 37676 (5292) | 33585 (1680) |
| Completeness (%) | 96.7 (94.6) | 92.2 (48.9) |
| Multiplicity | 8.7 (6.5) | 5.1 (1.6) |
| $R_{merge}$(%) | 6.2 (67.8) | 5.4 (45.7) |
| $<I/\delta(I)>$ | 17.0 (2.7) | 15.4 (1.6) |
| $R_{work}$(%) | 17.31 | 21.5 |
| $R_{free}$(%) | 19.46 | 24.0 |
| Rmsd bond lengths (Å) | 0.005 | 0.005 |
| Rmsd bond angles (°) | 0.794 | 0.836 |
| B-factors | 12.61 | 12.43 |
| Ramachandran plot residues in favored regions (%) | 97.66 | 95.86 |

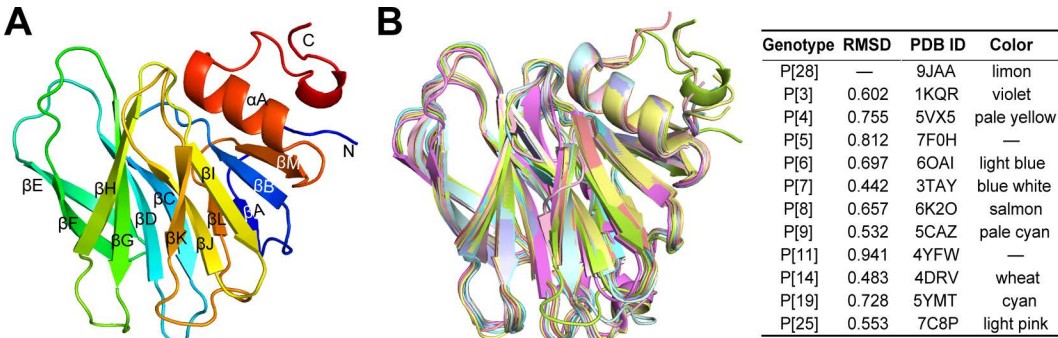

| Genotype | RMSD | PDB ID | Color |
|---|---|---|---|
| P[28] | — | 9JAA | limon |
| P[3] | 0.602 | 1KQR | violet |
| P[4] | 0.755 | 5VX5 | pale yellow |
| P[5] | 0.812 | 7F0H | — |
| P[6] | 0.697 | 6OAI | light blue |
| P[7] | 0.442 | 3TAY | blue white |
| P[8] | 0.657 | 6K2O | salmon |
| P[9] | 0.532 | 5CAZ | pale cyan |
| P[11] | 0.941 | 4YFW | — |
| P[14] | 0.483 | 4DRV | wheat |
| P[19] | 0.728 | 5YMT | cyan |
| P[25] | 0.553 | 7C8P | light pink |

**Fig 1. Structural analysis of P[28]-VP8*.** A. Crystal structure of P[28]-VP8* with two twisted antiparallel β-sheets. B. Superimposition of P[28]-VP8* structure to VP8*s of other genotypes. Values in the table represent root mean square deviations (RMSDs, unit: Å) of the Cα atoms of one VP8* monomer. VP8* structures were displayed in cartoon representation.

chain lengths. Microarray binding analysis of the P[28] RV-VP8* (Fig 2A) showed very strong signals to monovalent (Probe #18) and bivalent H-T1 (Probe #19), A-T1 (Probe #23) and B-T1 (probe #27). The strongest binding signal was observed to #19, due to the two identical H-T1 epitopes on two branches present in this probe. Interestingly, the same HBGAs on type 2 chains did not show any binding signals, nor the type 1 precursors of any chain lengths. It is not too surprising that A-T1-Tetra (Probe #22) did not exhibit any appreciable binding signal as the backbone is too short and the A epitope is too close to the amino-terminating linker of the probe.

To verify the binding specificity, on-array inhibition was carried out. The VP8* was pre-incubated with H-T1 pentasaccharide LNFP I at different concentrations before microarray overlay binding experiments. The percentage inhibition results (Fig 2B) showed declined binding signals to the H-T1-Penta and H-T1-Deca probes with increasing concentrations

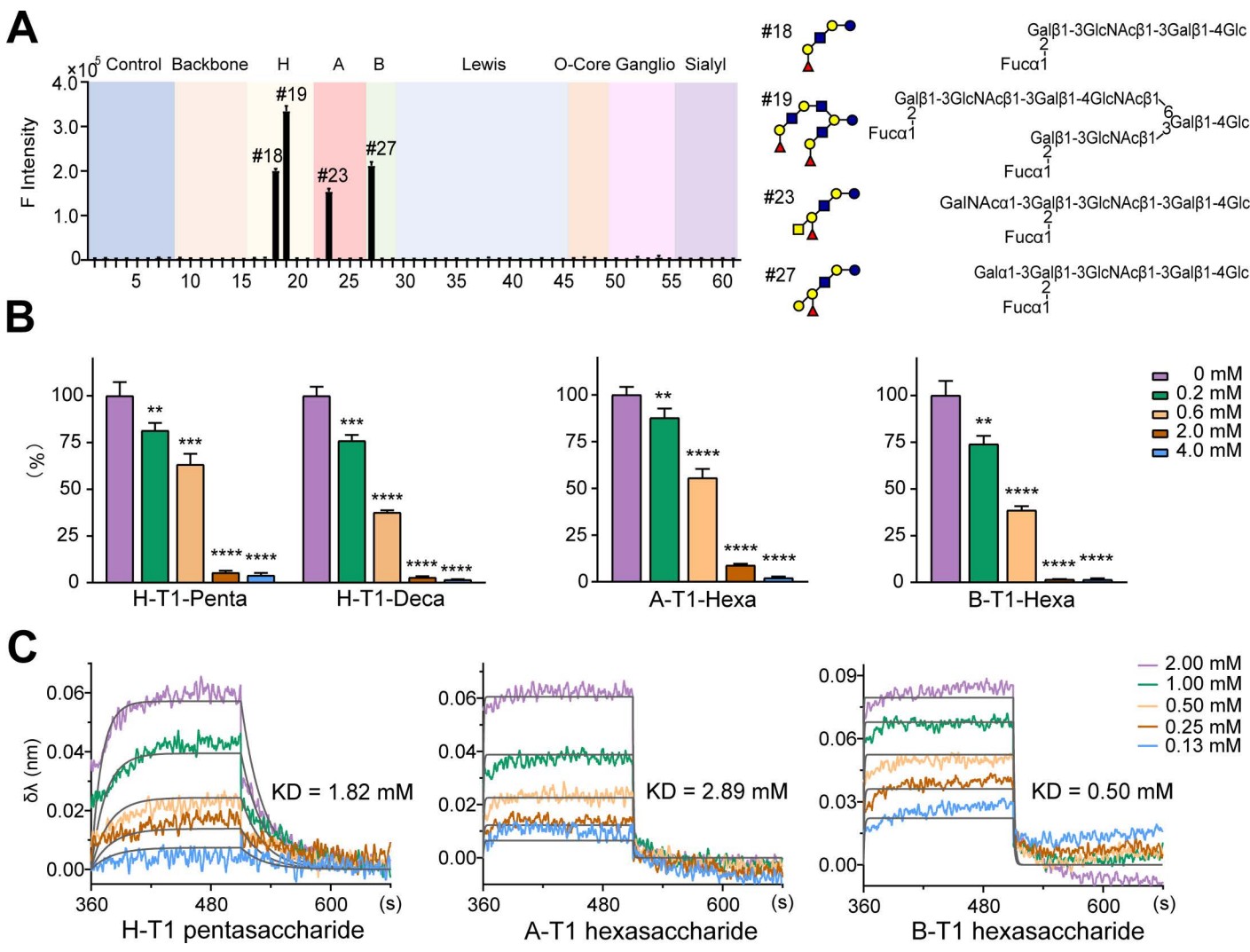

**Fig 2. Glycan receptors of P[28]-VP8* detected by glycan microarray and the binding kinetics analysis.** A. Binding signals of P[28]-VP8* on microarray. B. Binding signals of P[28]-VP8* pre-incubated with oligosaccharides at different concentration, ** P < 0.01, *** P < 0.001, **** P < 0.0001. C. BLI analysis of P[28]-VP8* with type 1 oligosaccharides.

of pre-incubated LNFP I. Similarly, the inhibitory effects can also be found for VP8* binding to A-T1-Hexa and B-T1-Hexa probes after pre-incubation with respective hexasaccharides (Fig 2B).

Bio-layer interferometry (BLI) was also carried out to assess the binding kinetics. As shown in Fig 2C, all three types 1 oligosaccharides displayed the same binding mode of association and dissociation at a fast rate. The $K_D$ values for VP8* binding to H-T1 pentasaccharide, A-T1 hexasaccharide, and B-T1 hexasaccharide were 1.82, 2.89 and 0.50 mM, respectively. Interestingly, the order of the $K_D$ values is identical to that of the intensities of microarray binding signals. The low mM $K_D$ values are consistent to the those of most carbohydrate-protein interactions [27].

### Determination of P[28]-VP8* binding site of H-T1 pentasaccharide by co-crystallization

To understand the structural basis of the interactions between P[28]-VP8* and the type 1 HBGAs, we selected H-T1 pentasaccharide, LNFP I, for co-crystallization with the VP8* protein as the type 1 blood group A and B epitopes share the

similar linear sequences LNFP I with an additional GalNAc and Gal residues, respectively. The structure of P[28]-VP8* in complex with LNFP I was determined at a resolution of 1.29 Å (Table 1). The bound ligand LNFP I showed clear electron density (Fig 3A). The edge part of the glycan binding site was composed of β-strand K, I-J loop, and 210-loop (residues 210213, which links the β-strand M and α-helix A), while the base part was formed by W81, M168, and Y175. The binding of H-T1 pentasaccharide involves a network of both direct hydrogen bonding interaction and stabilizing hydrophobic contact (Fig 3B and 3C). Apart from the reducing terminal Glc(1), all the other four monosaccharide residues Gal(2), GlcNAc(3), Gal(4) and Fuc(5) participate in the interactions. Gal(2) is in direct hydrogen bonding with H170 and N173. The main interactions are with GlcNAc(3), which formed two hydrogen bonds with E213 and one hydrogen bond with R210, in addition to hydrophobic interactions with W81, M168, Y175, and R210. Gal(4) participates in two hydrogen bonding interactions involving the side chain of R217 and also hydrophobic interactions with L185 and P186. The carbon atom C6 of the Fuc(5) is located within 4.0 Å of the side chain of R210 and also had a hydrophobic interaction with R210, consistent

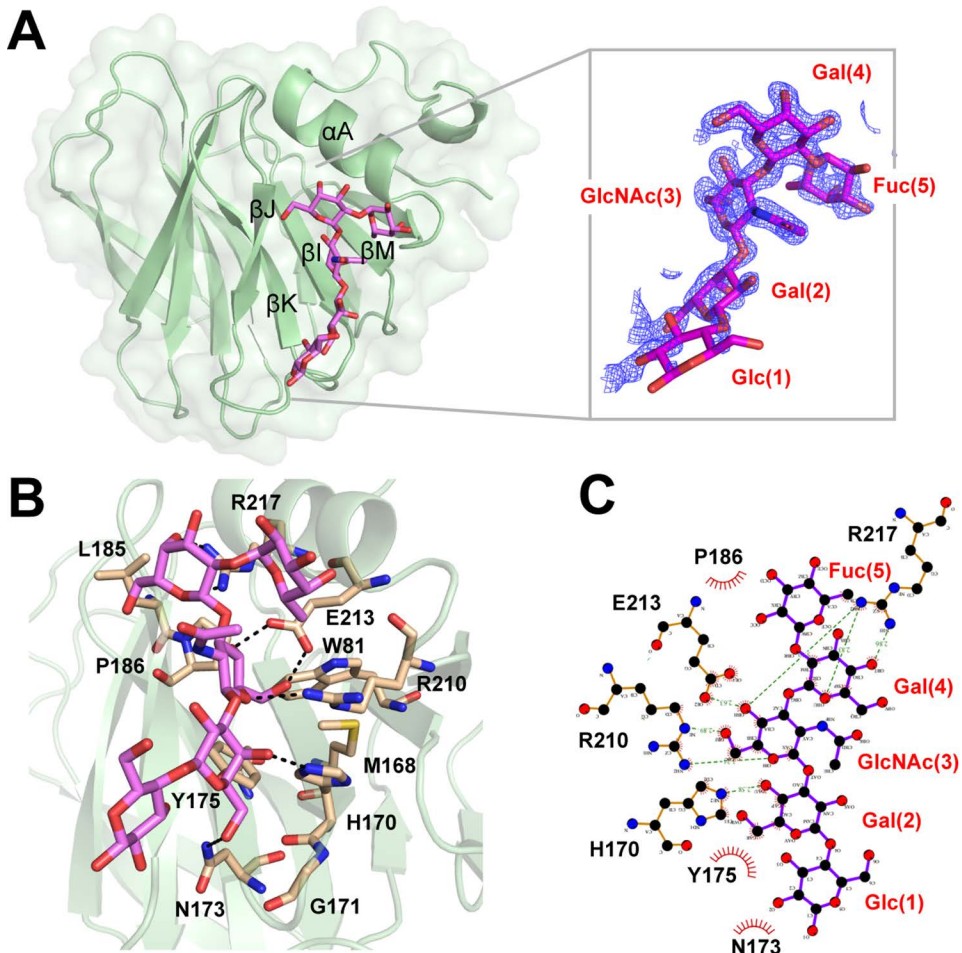

**Fig 3. Structural analysis of P[28]-VP8* in complex with H-T1 pentasaccharide LNFP I.** A. Cartoon representation of the P[28]-VP8*-LNFP I complex. P[28]-VP8* was shown in ribbon representation with LNFP I shown in stick. The electron density maps of LNFP I contoured at 1σ level are shown at the right panel. B. Detailed interactions among the amino acids of VP8*-LNFP I with the hydrogen bonds represented by dotted lines. C. Schematic diagrams of the interactions. The details of the interaction between LNFP I and P[28]-VP8* were calculated with Ligplot. Hydrogen bonds are indicated by dashed lines between the atoms involved, while hydrophobic contacts are represented by arcs with spokes radiating toward the ligand atoms they contact. Atoms colored black, red, and blue represent carbon, oxygen, and nitrogen, respectively.

with a similar result of that of P[6]/P[4] interaction with LNFP I reported previously [28]. Therefore, H170, R210, E213 and R217 contribute significantly to the interaction of VP8* with LNFP I.

### Determination of P[28]-VP8* binding sites of A-T1 and B-T1 hexasaccharides using on- array inhibition

Both A-T1 and B-T1 hexasaccharides share the same H-T1 pentasaccharide (LNFP I) backbone. To determine if A-T1 and B-T1 have similar binding site on VP8*, we used LNFP I to inhibit P[28]-VP8* binding to A-T1-Hexa (probe #23) and B-T1-Hexa (probe #27) on arrays. As shown in Fig 4A, the VP8* pre-incubated with LNFP I at different concentrations showed decreased binding signals to both A-T1-Hexa and B-T1-Hexa with increasing LNFP I concentrations, very similar to LNFP I inhibitory effect of VP8* binding to H-T1-Penta.

### Further analysis of P[28]-VP8* binding site of Type 1 blood group ABH antigens by molecular docking/dynamics simulations and site-specific mutagenesis

Molecular docking and dynamics simulations were also used to validate the above results and to obtain more information on the interactions. Docking of the VP8* with the H-T1 pentasaccharide LNFP I was generated first. Compared with co-crystallization, the docking provided similar structural inside of the interaction. As shown in Fig 4B, all the key amino acid residues involved in the interaction revealed in the co-crystallization were reproduced, with a docking score of -6.008. Docking analysis of A-T1/B-T1 hexasaccharide with VP8* was then carried out and negative docking scores of -5.740 and -6.636 for the interactions with A-T1 and B-T1, respectively, in the same grid box, indicating the accuracy of their interaction with the binding pocket. Both binding conformations of the two hexasaccharides in the active binding pocket involved hydrogen bond interactions.

The stability of the complexes was evaluated by monitoring the root-mean-square deviation (RMSD) of the protein and glycan Cα atoms (Fig 4C). All the three complexes were found to be stable within 100 ns of simulation. The RMSD of free protein increased slightly to 4 Å. For the complex of VP8*/H-T1 pentasaccharide, the RMSD value remained constant for the whole simulation process. For VP8*/A-T1 hexasaccharide, the highest fluctuation was observed with RMSD of 4 Å at 0–40 ns and remained stable for the rest of the process. For VP8*/B-T1 hexasaccharide, the RMSD value also remained constant for the entire simulation process. The dynamics simulations elucidated the conformational stability of the three complexes. R210, E213, and R217 of VP8* played predominant roles in the binding activity, basically consistent with co-crystallization results. Interestingly, the Q214, which was not found in the interaction of LNFP I and VP8* in co-crystallization, showed stronger hydrogen bonds with A-T1 hexasaccharide in the dynamics binding process.

Furthermore, we prepared five single-site mutants of VP8* with replacement of the following amino acid residues by alanine: H170A, R210A, E213A, Q214A and R217A, to verify the important residues for the interaction of VP8* with type 1 HBGAs. Apart from Q214A, all other mutants lost their binding activities on arrays (Figs 4D and S2 Fig), confirming that H170, R210, E213 and R217 are important for the binding. The mutant Q214A did not have any effect on H-T1 and B-T1 but only showed some effect on VP8* binding to A-T1-Hexa (Figs 4D and S2 Fig), which indicate that Q214 is a unique amino acid involved only in the binding to type 1 blood group A, consistent to the molecular simulation results described above.

### Comparison of glycan binding features of P[10] RV

In the P[I] genogroup, P[10] is the other RV which has been reported to infect humans. Using the dedicated glycan microarray, we analyzed the glycan receptors of human P[10]-VP8*. Our data showed prominent binding signals to HBGAs from the P[10]-VP8*, similar to P[28]-VP8*. Moreover, human P[10]-VP8* recognized additionally mucin O-glycan cores 2 and 4 (Fig 5A), consistent with a previous report [21]. The similarity of HBGA binding pattern indicated the genetically closely-related RVs may also exhibit antigenic similarity (Fig 5B).

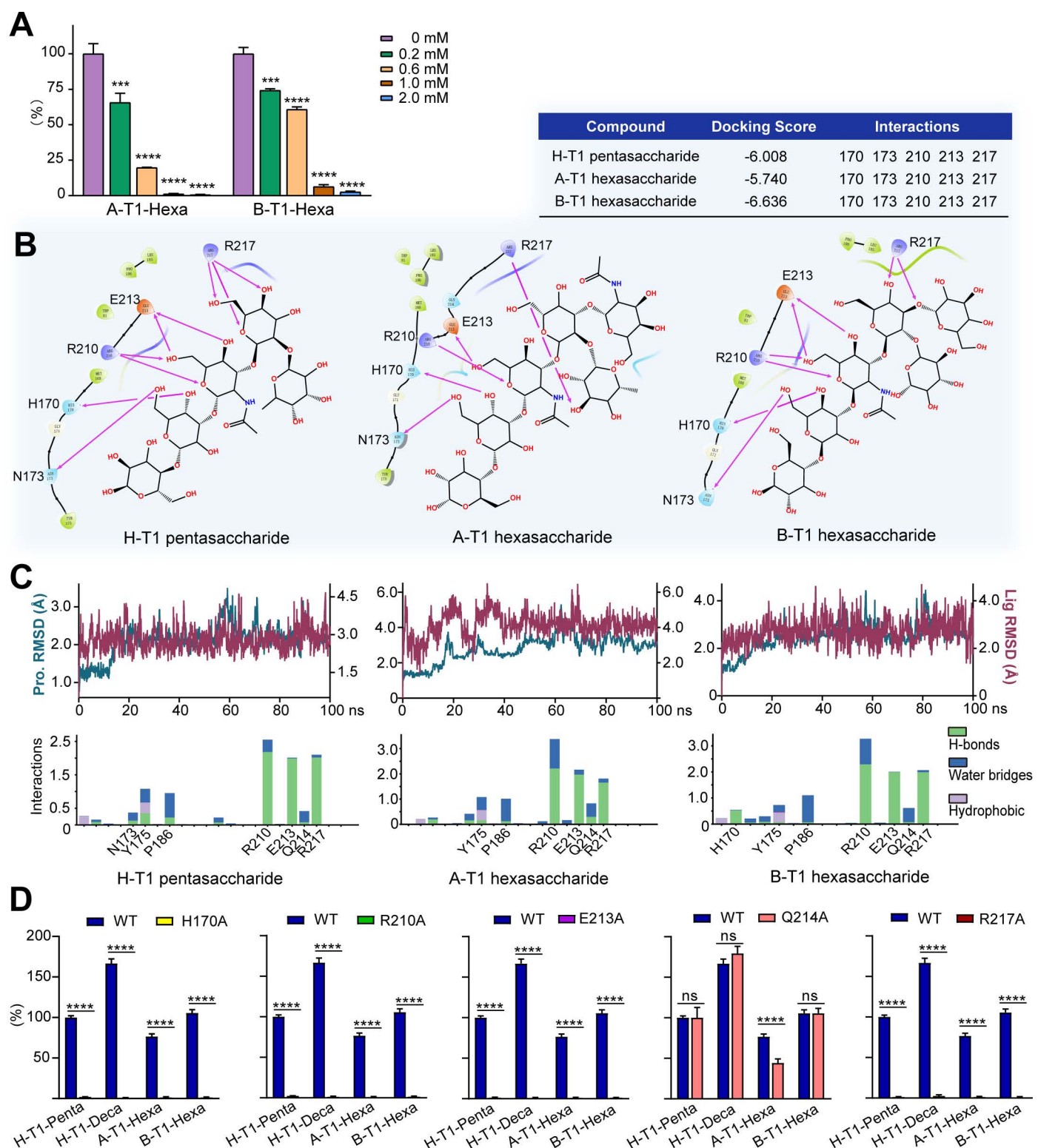

**Fig 4. Determination of VP8* binding site and the key amino acids with blood group ABH on type 1 chains.** A. On array inhibition of P[28]-VP8* binding to A-T1-Hexa and B-T1-Hexa using LNFP I as the inhibitor at different concentrations, *** P < 0.001, **** P < 0.0001. B. Molecular docking analysis of VP8* in complex with H-T1, A-T1, and B-T1 oligosaccharides. C. Molecular dynamics RMSD in complex with H-T1, A-T1, and B-T1 oligosaccharides. D. Microarray binding analysis of different P[28]-VP8* mutants in comparison with the wild-type, ns: P > 0.05, **** P < 0.0001.

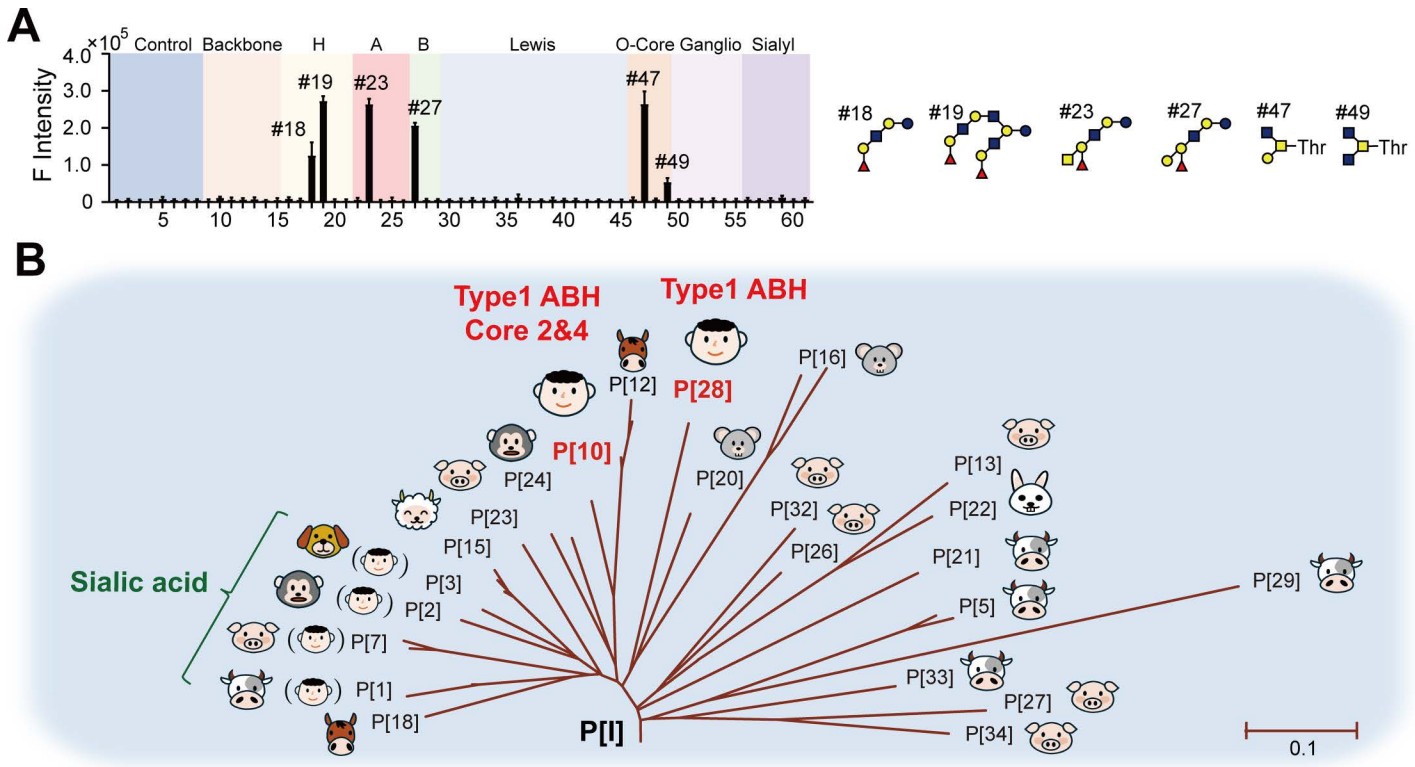

**Fig 5. Glycan binding properties of genetically closely-related RVs in the P[I] genogroup.** A. Microarray binding of human P[10]-VP8*. B. Phylogenetic analysis of P[I] genogroup RVs (modified from Fig 1 of Liu et al 2016 [29]).

## Discussion

Unrevealing RV interaction with host ligand is important for understanding of RV evolution, infection, and epidemic. There have been numerous reports on RV recognition of HBGAs [1,4–6,9], mucin O-glycan cores [7,21] and gangliosides [15–17,20] as potential attachment factors or receptors on the host cell surface. The mechanisms of many of these interactions have been elucidated mainly by crystal structures of VP8*s in complex with respective glycans, STD-NMR, and glycan microarrays. In the present work, a dedicated glycan microarray was constructed to contain all the different types of glycan sequences recognized by RV-VP8* as previously reported [13]. Using this microarray with a comprehensive panel of glycan probes we found that VP8* of P[28] RV of the P[I] group can bind specifically to blood group A, B, and H antigens on T1 chains. As the binding of P[28] RV to H-T1 is very similar to all the four genotype RVs, P[4], P[6], P[8] and P[19], in the P[II] genogroup, we further investigated the structural features of P[28] VP8* in relations to its binding properties and compared it with those of P[II] group viruses. From the structural comparison (Fig 6A), H-T1 pentasaccharide in the P[28]-VP8* complex is located at the same site as that in P[4]-, P[6]-, P[8]- and P[19]-VP8* (surface representation shown in Fig 6B). In a closer comparison with the most prevalent strain P[8]-VP8* (Fig 6C), the following four of the eleven amino acids in the glycan binding site are identical: W81, G171, R210, and E213. Although other amino acids are different, the orientation and space volume of the side chains are very similar.

The HBGAs recognized by P[28]-VP8* observed in this study are exclusively on T1 (Galβ1–3GlcNAc) rather than T2 (Galβ1–4GlcNAc) backbone chain. From the crystallographic analysis, the shallow groove formed by R210 and E213 has the major contribution to H-T1 pentasaccharide binding by forming the observed hydrogen bonds with GlcNAc, similar to

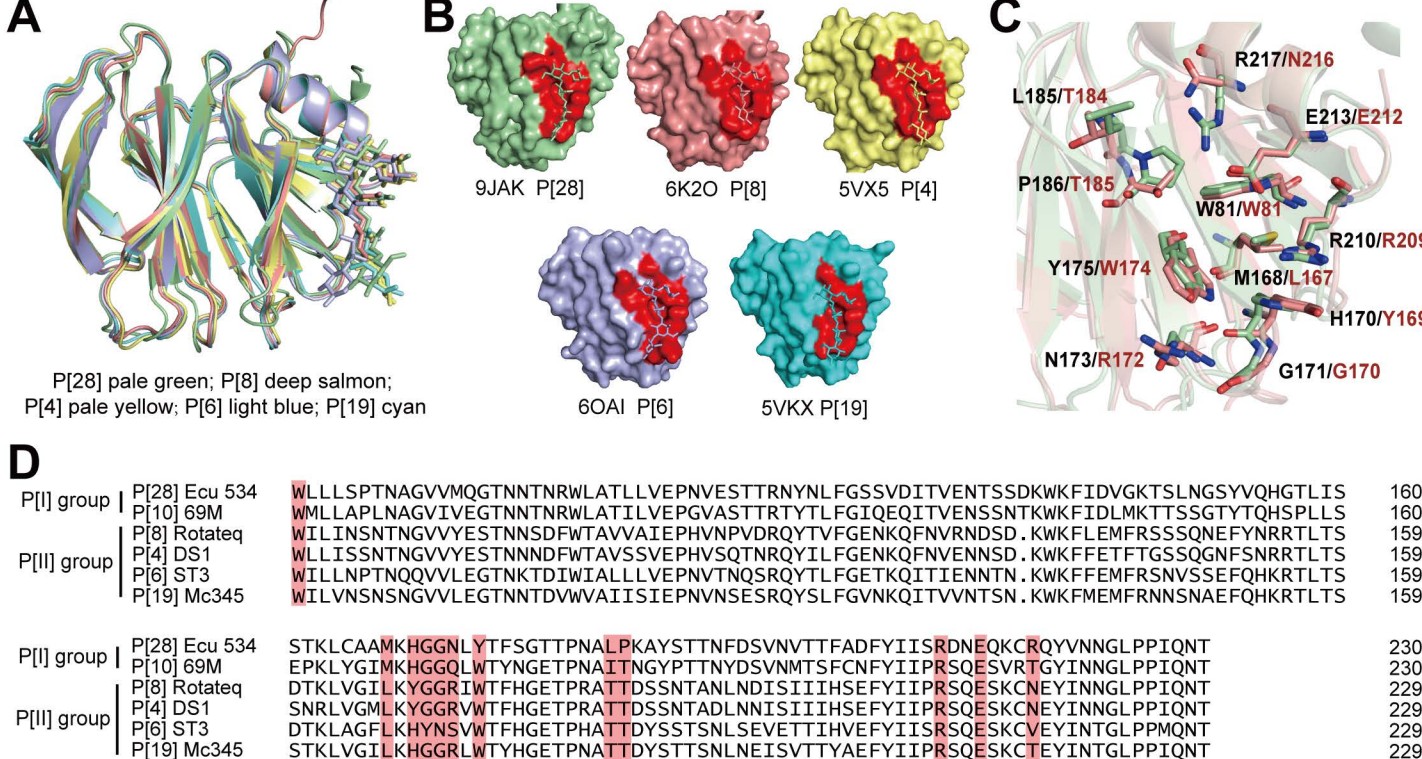

**Fig 6. Comparison of the glycan binding site of P[28]-VP8* with other RV-VP8*.** A. Comparation of the P[28] VP8*-LNFP I complex with published P[II] RV VP8* complexes. B. Surface representation of VP8*-LNFP I complex of P[28] and published VP8* complexes of RVs in the P[II] genogroup (the binding sites are in red). C. Comparison of the key amino acids of P[28]-VP8* in complex with LNFP I (pale green) with those of the P[8]-VP8* complex (deep salmon). D. Structure-based sequence alignment of VP8*s from selected P genotypes containing conserved amino acid residues (red) for binding to type 1 ABH (constructed using DNAMAN).

the P[8]-VP8*-LNFP I complex, in which R209 and E212 (Fig 6C and 6D) play the important role to the VP8* interaction with GlcNAc [3,30]. However, R217 of P[28]-VP8* has direct hydrogen bonding with the nonreducing Gal residue of LNFP I, and therefore is the key amino acid for selective binding to T1 backbone chain, whereas in the case of P[4]-, P[6]-, and P[8]-VP8*s of the P[II] group RVs, the VP8* interaction with the Gal is through E212 [28]. These amino acids are considered to be the key residues for VP8* recognition of T1 backbone chain. The role of fucose residue in the H antigen interacting with VP8* has been different for different human RVs. In the present case, both P[28] and P[10] recognize fucosylated glycans including the ABH antigens but do not bind to the non-fucosylated backbone chains. As described in the Results, the co-crystal structure of the complex LNFP I-P[28]-VP8* clearly showed a hydrophobic interaction between Fuc-C6 and R210 side chain.

There are only two P genotypes in the P[I] group that infect humans. We now carefully compared the binding characteristics of P[10] and P[28] using the same microarray containing a comprehensive panel of glycan probes that have been reported as potential ligands of rotaviral VP8*. The results are consistent to the saliva binding of P[10] RV VP8* shown in the previous report [21], but we further clarified the ABH(O) antigens on T1 chain only. However, the results of the microarray binding are different from the glycan binding reported previously using microwells [21]. The difference in glycan binding is probably due to the difference in glycan probe density, probe linker composition, and glycan presentation on

solid surfaces [31], such as different microarray slides and microwell surfaces. Therefore, for direct comparison of glycan binding specificities, it would be important to use a similar platform.

In the present study, we investigated glycan binding property of the VP8* protein of human P[28] RV and found that it can bind blood group A, B, and H antigens on type 1 chain only but do not recognize any Lewis epitopes or mucin O-glycan cores. This specificity is different from those of most human RVs. Although all the prevalent human RVs (P[8], P[4] and P[6]) can bind H-T1, none of them recognize the B antigens. P[8], P[4] and P[6] can additionally bind to Lewis b [4,6,32]. P[4] and P[6] also bind to the A antigen [33], while P[6] can recognize internal Lewis x [34] and mucin O-glycan core 2 [7]. To define the cell surface glycan receptors of the blood group ABH(O) and Lewis types is important not only for better understanding of the host susceptibility to different rotavirus infections mediated by their different Lewis phenotypes and secretor status, it may also help to explain the different efficacies of different rotavirus vaccines developed recently [35].

We consider two limitations in our present work. As P[28] is not a prevalent genotype with only limited case report, at present it is not possible to provide epidemiological analysis data. The present work has been based on the P[28]-VP8* recombinant proteins, lacking virus infection at the cellular level. Additional work to assess the features of glycan-RV interactions using rescued recombinant viruses with site mutations will strengthen the conclusion obtained here when the very challenging technical hurdle of reverse genetics-based virus rescue for the P genotypes can be overcome [36].

P[28] RVs have been considered to infect humans exclusively, but a recent genetic analysis [20] indicated that a human P[28] RV strain could represent a zoonotic infection between bat and humans. Therefore, the possibility of future P[I] genogroup RV epidemic could not be ignored, and the RV surveillance might need to take P[I] genogroup RVs into account in the future.

## Materials and methods

### Protein expression and purification

The wild-type P[28]-VP8* and its mutants (H170A, R210A, E213A, Q214A, R217A) gene segments (GenBank: EU805773.1) were synthesized by GenScript Biotech Corp. (Nanjing, China). The full VP8* gene (encoding amino acids (aa) 1–232, including five single site mutants with the replacement of Alanine) and partial VP8* gene (encoding 62–232 aa) were cloned into expression vector pGEX4T-1 respectively, with an N-terminal glutathione S-transferase GST tag and a thrombin cleavage site of each. The recombinant GST-VP8* and the mutants were all expressed in *E. coli* BL21 (DE3) and purified by Glutathione Sepharose 4B (GE healthcare, Delaware, USA). For crystallization, the GST tag of VP8*(62–232 aa) was cleaved by thrombin before reloading the protein mixtures onto a Glutathione Sepharose column to remove the released GST tag. The purified proteins were analyzed by SDS-PAGE (S1A Fig) and the protein concentration were determined by BCA protein assay kits (Beyotime Biotechnology, Shanghai, China).

### Crystallization, data collection and processing

Crystallization conditions for P[28]-VP8* (12.0 mg/mL) were screened by hanging-drop vapour diffusion method with Wizard Classic Crystallization Screen series kits (Rigaku, Tokyo, Japan) at 16°C. The crystals formed under the condition (25.0 mmol/L malonic acid, 37.5 mmol/L imidazole, 37.5 mmol/L boric acid, pH 5.0, 30% poly ethylene glycol (PEG) 1500) and were harvested with the screen condition containing 25% glycerol. To obtain crystals of VP8*-HBGA complex, VP8* was co-crystallized with LNFP I, with a molar ratio of 1:100, under the condition (100.0 mmol/L KBr, 35% PEG methyl ether 2000) at 16°C. X-ray diffraction data for both unliganded and liganded VP8* crystals were collected on BL10U2 station of Shanghai Synchrotron Radiation Facility [37]. These data were processed with Aquarium data-processing and experiment information management system [38]. The molecular-replacement were based upon amino-acid sequence alignment with human P[14] VP8* structure (PDB ID: 4DS0) and then the data were refined using PHENIX suite. The oligosaccharide moiety of LNFP I was generated using PRODRG program and modeled into the electron density using

PLOS Pathogens

COOT and validated by computing simulated annealing omit maps using PHENIX suite. Data collection and refinement statistics are given in Table 1. Root mean square deviation (RMSD) values of the matching Cα atoms between P[28]-VP8* and other representative VP8* structures were calculated by PyMOL (https://pymol.org/2/). Ligand interactions were firstly searched by CCP4 (http://ccp4.ac.uk/) with the distance of less than 5 Å and verified by Ligplot (https://www.ebi.ac.uk/thornton-srv/software/LIGPLOT/) and PyMOL.

## Glycan microarray analysis and on-array inhibition assay

Glycan microarrays used in this study contain 58 glycan probes purified from human milk or synthesized chemically, consisting of blood group A, B, H, Le$^a$, Le$^b$, Le$^x$, and Le$^y$, mucin O-glycan cores 1–4 and different ganglio-oligosaccharides (S1 Table). The binding assays were performed following the previous protocol [39]. Briefly, the slides were blocked with the blocking solution (1% BSA in Tris-buffered saline with 0.1% Tween 20) at 25°C for 1 h, washed, and dried. The VP8* and mutants with GST tag were diluted in the blocking solution to achieve a final concentration of 100 µg/mL. Then, the dilutions were incubated with the slides for 1.5 h. After being washed and dried, the slides were finally incubated with AlexaFluor-647-conjugated anti-GST tag antibody for 1 h at 25°C. For inhibition assay, the proteins were firstly incubated with oligosaccharides at 25°C for 20 min. Then the protein-glycan mixtures were incubated with the BSA blocked slides and performed the same subsequent operation. The binding signals were detected by a GenePix 4300A fluorescence scanner and quantified using GenePix Pro Software 7.3 (Molecular Devices, Union City, USA).

## Binding kinetics analysis

The binding kinetics analysis between P[28]-VP8* and various oligosaccharides was carried out on an Octet RED K2 instrument (ForteBio, Fremont, USA). P[28]-VP8* was biotinylated according to the instructions of Biotinylation Kit (BMD labservice, Suzhou, China) before use. Double reference experiments with super streptavidin (SSA) biosensors were utilized. H-T1 pentasaccharide, A-T1 and B-T1 hexasaccharides were purchased from Elicityl (Grenoble, France). The oligosaccharides were dissolved and diluted into 125–2000 µmol/mL using PBST (phosphate buffered saline, pH7.2 with 0.05% Tween-20). The kinetics analysis mainly involves three steps: placing the sensors into 50 µg/mL biotinylated P[28]-VP8* (loading, for 300 s), transferring the sensors into oligosaccharides at different concentration (association, for 120s), and transferring the sensors into buffer (dissociation, 120s). The kinetic parameters were analyzed using ForteBio Data Analysis Software 8.0 (ForteBio, Fremont, USA).

## Molecular docking and dynamics simulations

Potential interaction of P[28]-VP8* with type 1 oligosaccharides was investigated using the molecular modeling based on the Schrödinger software suite 2022–1 (Schrödinger, New York, USA). Protein Preparation Wizard tool in Schrödinger suite was used to optimize the VP8* (PDB ID: 9JAA) by removing water and adding the missing hydrogen atoms and bond orders. PROPKA program was used to calculate the ionization state of polar amino acids at pH 7.0. A full-atom energy minimization of the protein structure was carried out using the OPLS4 force field to refine the geometry structure improve force field accuracy and the previous H-bond network by minimizing steric clashes. LNFP I was prepared from the co-crystal (PDB ID: 5VX9) and A/B type 1 hexasaccharides were generated from LNFP I using R-Group enumerate program, followed by the generation of correct ionizable state at pH 7.0. As for the receptor grids, the grid of VP8* was set as 29 Å × 29 Å × 29 Å. Docking calculations were completed with Glide standard precision docking protocol and 20 poses of complexes were exported. Some top-ranking poses were checked manually, the optimal poses were analyzed and exported with Maestro.

Molecular dynamics simulation was performed using the Desmond suite for the VP8* and T1 oligosaccharide complex. All the complexes were solvated individually by placing them in an explicit water box of size 10 Å with a single-point charge water model TIP3P with periodic boundary condition. The OPLS4 force field was used to minimize the energy of

the entire solvated system. After minimization, a short NPT ensemble equilibration simulation was performed to relax the model before 100 ns production-quality molecular dynamics simulation. The system was slowly heated to maintain a temperature of 300 K and pressure by using the Nose-Hoover thermostatic algorithm and the Martina-Tobias-Klein method. Particle-Mesh Ewald method was utilized to calculate long-range electrostatic interactions keeping a grid spacing of 0.8 Å. When the simulation was completed, the Simulation Interaction Diagram tool implemented in the Desmond package was utilized to analyze the interactions.

## Supporting information

**S1 Fig.** A. SDS-PAGE analysis of expressed P[28]-VP8* and its mutants (GST-labelled VP8* at ~50 kDa and the impurity GST tag at ~26 kDa) . B. Demonstration of the GST impurity does not have binding activity to any of the glycan probes used in the microarray.
(TIF)

**S2 Fig.** Microarray binding signals of different P[28]-VP8* mutants in comparison with the wild-type.
(TIF)

**S1 Table.** List of glycan probes in microarrays.
(DOCX)

## Acknowledgments

We acknowledge the staff of beamline BL10U2 at the Shanghai Synchrotron Radiation Facility for assistance during the data collection (https://cstr.cn/31124.02.SSRF.BL10U2).

## Author contributions

**Conceptualization:** Wengang Chai, Zhaojun Duan, Jingyu Yan.

**Data curation:** Yi Zheng, Xiaoman Sun, Yuting Li, Beibei Huang, Cuiyan Cao.

**Formal analysis:** Xiaoman Sun, Yang Chen, Han Zhou, Cuiyan Cao, Wengang Chai.

**Funding acquisition:** Yi Zheng, Wengang Chai, Jingyu Yan, Xinmiao Liang.

**Investigation:** Yi Zheng, Yuting Li, Beibei Huang.

**Methodology:** Xiaoman Sun, Yuting Li, Yang Chen.

**Project administration:** Wengang Chai, Zhaojun Duan, Dandi Li, Jingyu Yan, Xinmiao Liang.

**Resources:** Xiaoman Sun, Dandi Li, Jingyu Yan.

**Supervision:** Dandi Li, Jingyu Yan, Xinmiao Liang.

**Validation:** Yang Chen, Han Zhou.

**Visualization:** Yi Zheng, Xiaoman Sun.

**Writing – original draft:** Yi Zheng, Xiaoman Sun, Jingyu Yan.

**Writing – review & editing:** Wengang Chai, Dandi Li, Xinmiao Liang.

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
