## [Decision Letter · Decision Letter 0]

Specific Binding of Human P[28] Rotavirus VP8* Protein to Blood Group ABH Antigens on Type 1 Chains

PLOS Pathogens

Dear Dr. Yan,

Thank you for submitting your manuscript to PLOS Pathogens. After careful consideration, we feel that it has merit but does not fully meet PLOS Pathogens's publication criteria as it currently stands. Therefore, we invite you to submit a revised version of the manuscript that addresses the points raised during the review process.

Please submit your revised manuscript within 60 days Apr 22 2025 11:59PM. If you will need more time than this to complete your revisions, please reply to this message or contact the journal office at plospathogens@plos.org. Please include the following items when submitting your revised manuscript:

We look forward to receiving your revised manuscript.

Kind regards,

Carlos F. Arias, Ph.D.

Guest Editor

PLOS Pathogens

Alexander Gorbalenya

Section Editor

PLOS Pathogens

Editor-in-Chief

PLOS Pathogens

orcid.org/0000-0003-2946-9497

Editor-in-Chief

PLOS Pathogens

orcid.org/0000-0002-7699-2064

**Additional Editor Comments:**

The reviewers recognize the ample array of different methodologies employed to investigate the glycan-binding specificity of the rotavirus P[28] VP8 protein. They also agree that the conclusions reached are supported by the data presented. However, two of the reviewers, with whom I agree, emphasize the need for functional validation of the findings.. Specifically, mutations in the four key amino acids identified for the VP8-glycan recognition should be introduced into the virus using the available reverse genetics systems and the infectivity and binding capacity of the rescued viruses assessed. Additionally, the effect of preincubation of selected glycans on the infectivity and binding of P[28] rotavirus also needs to be evaluated.

**Journal Requirements:**

At this stage, the following Authors/Authors require contributions: Yi Zheng, Xiaoman Sun, Yuting Li, Beibei Huang, Yang Chen, Han Zhou, Cuiyan Cao, Wengang Chai, Zhaojun Duan, Dandi Li, Jingyu Yan, and Xinmiao Liang. Please ensure that the full contributions of each author are acknowledged in the "Add/Edit/Remove Authors" section of our submission form.

2) Some material included in your submission may be copyrighted. According to PLOSu2019s copyright policy, authors who use figures or other material (e.g., graphics, clipart, maps) from another author or copyright holder must demonstrate or obtain permission to publish this material under the Creative Commons Attribution 4.0 International (CC BY 4.0) License used by PLOS journals. Please closely review the details of PLOSu2019s copyright requirements here: PLOS Licenses and Copyright. If you need to request permissions from a copyright holder, you may use PLOS's Copyright Content Permission form.

Potential Copyright Issues:

- Figure 5B. Please confirm whether you drew the images / clip-art within the figure panels by hand. If you did not draw the images, please provide a link to the source of the images or icons and their license / terms of use; or written permission from the copyright holder to publish the images or icons under our CC BY 4.0 license. Alternatively, you may replace the images with open source alternatives. See these open source resources you may use to replace images / clip-art:

3) Please ensure that the funders and grant numbers match between the Financial Disclosure field and the Funding Information tab in your submission form. Note that the funders must be provided in the same order in both places as well.

**Reviewers' Comments:**

Reviewer's Responses to Questions

**Part I - Summary**

Reviewer #1: The article presented by Zheng and collaborators describes the interaction of a human rotavirus VP8* belonging to the P[28] genotype with specific histo-blood group antigens (HBGAs). The interaction of human rotaviruses with glycans, such as the HBGAs, has win relevance in the last years due to its implication in host species restriction and genetic susceptibility to infections in humans amongst others.

The authors have focused their attention to a rare human genotype P[28] that is closely related to bat rotaviruses and link the putative cross-species jump to the different ability of this genotype to bind most relevant human HBGAs as they are the H antigen, the A blood group and the B blood group.

The reviewer believes that the methodology used is appropriated to obtain the presented results and that the conclusions are aligned with the results obtained. Several technologies were successfully applied including glycan microarrays, crystallography, biolayer interferometry and site directed mutagenesis all of them confirming the results presented by the authors.

Reviewer #2: The biochemical data seems solid. However, there is no virological data to validate the findings. Without it, it is hard to interpret how significant the results are. For example, the VP4 mutations outlined in Fig. 4D cannot be validated without functional analysis.

Reviewer #3: This study investigated the glycan-binding specificity of the rotavirus P[28] VP8 protein with respect to type 1 chain ABH antigens, elucidating its unique interaction mechanism. By integrating X-ray crystallography, glycan microarray analysis using a dedicated probe library, bio-layer interferometry, site-specific mutagenesis, and molecular docking and dynamics simulations, it was revealed that the P[28] VP8 exhibits broad HBGA binding, which is distinct from the prevalent human rotavirus strains P[8], P[4], and P[6]. These findings provide experimental evidence for interspecies strain reassortment within the P[I] genogroup and highlight the need for enhanced surveillance to address potential cross-species transmission risks associated with P[I] rotaviruses.

**Part II – Major Issues: Key Experiments Required for Acceptance**

Reviewer #1: This reviewer has not found major issues in this article

Reviewer #2: While the genetic manipulation of host cells to remove specific glycans would be challenging, with the current reverse genetics system, the authors should be able to rescue recombinant rotaviruses (monoreassortant viruses in SA11 background for instance) carrying these mutations and test how the infectivity is affected.

Reviewer #3: 1. This study primarily investigates the P[28] genotype, need to augment data on the prevalence of P[28] genotype viruses among human rotaviruses to underscore its epidemiological significance.

2. The study's conclusions are largely based on the P[28]-VP8* recombinant protein. Although the in vitro experiments are thorough and detailed, they lack validation at the cellular level. For instance, assessing the change in infection titer after pre-incubating glycans such as H-T1-Penta with the P[28] genotype rotavirus before infecting cells is essential�it is necessary to demonstrate whether HBGA is a key receptor in the infection process of P[28] rotavirus.

3. Figure 2B evaluated four glycans (H-T1-Penta, H-T1-Deca, A-T1-Hexa, and B-T1-Hexa); however, only three were further analyzed for their interactions with VP8* in Figure 2C and subsequent experiments. This discrepancy requires clarification to ensure completeness of the dataset.

4. Figure 4D confirmed four key amino acid sites. If the rotavirus with site mutations can be rescued using a reverse genetics system and the effects of these mutations on the virus's infectivity and binding activity with various glycans are further analyzed, the conclusions of this study might be strengthened.

**Part III – Minor Issues: Editorial and Data Presentation Modifications**

Reviewer #1: Although the reviewer does not appreciate any major concerns there are some minor issues that should be improved:

Line 66. The authors state that the human P[11] genotype recognizes both type I and type II precursors, but the articles cited show that this genotype binds type II precursors (Galb1-4GlcNAc) but not type I (Galb1-3GlcNAc).

Line 68. The authors state that there are at least 38 P genotypes in group A rotavirus, but this number has increased dramatically since 2020, currently there are at least 58 P genotypes (https://rega.kuleuven.be/cev/viralmetagenomics/virus-classification/rcwg). Please update the number and refeer to the website that keeps the rotavirus genotype updated.

Line 136. The authors found that, contrary to other human rotaviruses, the P[28] VP8* utilized in this study was not able to bind the precursors. This fact should be further discussed in the discussion section. Please notice that for instance in the P[8] genotype interactions with HBGAs the fucose has not direct interactions with the protein (Gozalbo-Rovira et al., 2019) wile the results presented here show direct interactions of the fucose wit the protein.

Line 145. In this paragraph the authors describe their resuts obtained by biolayer interfrerometry but it is not clear here, nor in the matherial and methods section (line 373), the source and the nature of the oligosacchares used, please provide that information.

Line 154. In the text to figure 2B it is not clear wich sugar was used bo block the binding in each case. Please improve the description of this panel.

Line 281. There are mistakes in the T1 and T2 descriptionsT1 shlould be Galb1-3GlcNac and T2 Galb1-4GlcNac. Please correct

Reviewer #2: Statistics are missing from all the panels (Fig. 2B, 4A, and 4D).

Reviewer #3: 1. In the background section, the authors propose that the P[28] strain may have originated from reassortment between human and bat rotaviruses. Further explanation is needed on which genes may have experienced reassortment. Given this study's focus on the VP8* segment, the potential reassortment of the VP4 fragment is pertinent to the research's significance.

2. The introduction section needs additional cited references, such as in lines 58-59: "The P genotypes are further classified into five genogroups [1], P[I] to P[V], based on the sequences of the spike protein VP8*, one of the two proteolytic fragments of VP4." And in lines 73-74: "The glycan bindings of animal strains of P[1], P[2], P[3], and P[7] are reported to be sialic acid-dependent."

3. In line 68, the authors state, "There have been at least 38 P genotypes identified until 2020." However, existing research has reported more than 50 P genotype rotaviruses, so this description needs to be updated.

4. Please discuss why the structural differences between P[23] and other P[I] types, such as P[5]-VP8, are more pronounced than those in other types, like P[9] and P[14]-VP8 (Figure 1B).

5. In Figure S1, each VP8* protein exhibits moderate purity. Notably, the GST-fused VP8 proteins display significant impurity bands between 31-43 kDa. A discussion is warranted to determine whether these impurities might impact the relevant experimental results.

6. Please discuss the limitations of the article.

7. In the last sentence of the discussion section, it is mentioned that the monitoring of the P[23] strain should not be neglected. Is this statement too absolute? Although the monitoring of the P[23] genotype has certain epidemiological value, more data on circulating viruses is still needed to support it.

PLOS authors have the option to publish the peer review history of their article (what does this mean? ). If published, this will include your full peer review and any attached files.

**Do you want your identity to be public for this peer review?** For information about this choice, including consent withdrawal, please see our Privacy Policy .

Reviewer #1: No

Reviewer #2: No

Reviewer #3: No

**Figure resubmission:**

**Reproducibility:**



---

## [Decision Letter · Decision Letter 1]

Dear Prof. Yan,

We are pleased to inform you that your manuscript 'Specific Binding of Human P[28] Rotavirus VP8* Protein to Blood Group ABH Antigens on Type 1 Chains' has been provisionally accepted for publication in PLOS Pathogens.

Best regards,

Carlos F. Arias, Ph.D.

Guest Editor

PLOS Pathogens

Alexander Gorbalenya

Section Editor

PLOS Pathogens

Sumita Bhaduri-McIntosh

Editor-in-Chief

PLOS Pathogens

orcid.org/0000-0003-2946-9497

Michael Malim

Editor-in-Chief

PLOS Pathogens

orcid.org/0000-0002-7699-2064

Reviewer Comments (if any, and for reference):

Reviewer's Responses to Questions

**Part I - Summary**

Reviewer #2: I am satisfactory with the revised version of the manuscript.

Reviewer #3: (No Response)

**Part II – Major Issues: Key Experiments Required for Acceptance**

Reviewer #2: (No Response)

Reviewer #3: In the response, the authors mentioned difficulties encountered when trying to insert VP4-encoding genes using SA11 and LLR as backbones to rescue recombinant P genotype rotaviruses. We understand this challenge, but since the authors' team has the genomic sequence of the P[28] strain, could you establish a reverse genetics system based on the P[28] strain's own sequence to rescue the P[28] strain? Additionally, could you verify at the cellular level whether HBGA is a critical receptor in the infection process of the P[28] rotavirus? Evaluating changes in infection titers, such as pre-incubating glycans (such as H-T1-Penta) with the P[28] genotype rotavirus before cell infection, would support the study and enhance its value.

**Part III – Minor Issues: Editorial and Data Presentation Modifications**

Reviewer #2: (No Response)

Reviewer #3: (No Response)

PLOS authors have the option to publish the peer review history of their article (what does this mean? ). If published, this will include your full peer review and any attached files.

**Do you want your identity to be public for this peer review?** For information about this choice, including consent withdrawal, please see our Privacy Policy .

Reviewer #2: No

Reviewer #3: No

---

## [Editor Report · Acceptance letter]

Dear Prof. Yan,

We are delighted to inform you that your manuscript, "Specific Binding of Human P[28] Rotavirus VP8* Protein to Blood Group ABH Antigens on Type 1 Chains," has been formally accepted for publication in PLOS Pathogens.

Best regards,

Sumita Bhaduri-McIntosh

Editor-in-Chief

PLOS Pathogens

orcid.org/0000-0003-2946-9497

Michael Malim

Editor-in-Chief

PLOS Pathogens

orcid.org/0000-0002-7699-2064